# Efficient Catalytic Synthesis of Condensed Isoxazole Derivatives via Intramolecular Oxidative Cycloaddition of Aldoximes

**DOI:** 10.3390/molecules27123860

**Published:** 2022-06-16

**Authors:** Irina A. Mironova, Valentine G. Nenajdenko, Pavel S. Postnikov, Akio Saito, Mekhman S. Yusubov, Akira Yoshimura

**Affiliations:** 1Research School of Chemistry and Applied Biomedical Sciences, Tomsk Polytechnic University, 634050 Tomsk, Russia; irina190793@mail.ru (I.A.M.); pavelpostnikov@gmail.com (P.S.P.); 2Department of Chemistry, M. V. Lomonosov Moscow State University, 119991 Moscow, Russia; nenajdenko@gmail.com; 3Division of Applied Chemistry, Institute of Engineering, Tokyo University of Agriculture and Technology, Tokyo 184-8588, Japan; akio-sai@cc.tuat.ac.jp; 4Faculty of Pharmaceutical Sciences, Aomori University, 2-3-1 Kobata, Aomori 030-0943, Japan

**Keywords:** hypervalent iodine, hydroxy(aryl)iodonium, catalysis, aldoximes, intramolecular cycloaddition, nitrogen heterocycles

## Abstract

The intramolecular oxidative cycloaddition reaction of alkyne- or alkene-tethered aldoximes was catalyzed efficiently by hypervalent iodine(III) species to afford the corresponding polycyclic isoxazole derivatives in up to a 94% yield. The structure of the prepared products was confirmed by various methods, including X-ray crystallography. Mechanistic study demonstrated the crucial role of hydroxy(aryl)iodonium tosylate as a precatalyst, which is generated from 2-iodobenzoic acid and *m*-chloroperoxybenzoic acid in the presence of a catalytic amount of *p*-toluenesulfonic acid.

## 1. Introduction

Heterocycles play a key role in modern drug discovery and agrochemistry [1,2,3,4,5,6]. Heterocyclic fragments can be found in the structure of many marketed small molecules. Currently, approximately 60% of approved US FDA drugs are derivatives of nitrogen heterocycles [7,8]. Isoxazole fragment is among the most popular heterocyclic fragments of drugs. These heterocycles have two connected heteroatoms in the structure. As a result, isoxazoles can form specific interactions with various protein targets via hydrogen bonds, as well as stacking and hydrophilic interactions. All these structural advantages have made them very popular in drug discovery. Their derivatives exhibit a broad range of bioactivities, such as being anticancer, antibacterial, antifungal, antimicrobial, antiviral, and antituberculosis [9,10,11,12,13,14,15].

The 1,3-dipole cycloaddition reaction is one of the most powerful methods to construct five-membered heterocycles [16,17,18,19,20,21,22,23,24,25,26]. The cycloaddition of nitrile oxides with alkenes and acetylenes is often used in the synthesis of isoxazoles and isoxazolines [27,28,29,30,31,32,33,34,35,36,37]. However, nitrile oxides are unstable species usually generated in situ from aldoximes under appropriate conditions [38,39,40]. The intramolecular version of the cycloaddition of nitrile oxides with alkenes and acetylenes is less investigated. On the other hand, this approach can provide an efficient approach to condensed heterocyclic systems containing isoxazole or isoxazoline rings.

This study is devoted to the investigation of synthetic approaches to isoxazole- or isoxazoline-fused heterocycles via the catalytic intramolecular cycloaddition of alkyne- or alkene-tethered aldoximes using hypervalent hydroxy(aryl)iodonium species generated in the reaction system (Figure 1c), as well as the study of the reaction mechanism. Hypervalent iodine compounds are known as low-toxic, environmentally benign reagents that have been applied to various organic synthetic reactions [41,42,43,44,45,46,47,48,49,50,51,52,53,54,55,56,57,58,59,60,61,62,63,64]. In recent years, several examples of the oxidative cycloaddition of aldoximes with alkenes or alkynes were demonstrated using hypervalent iodine(III) species as oxidants (Figure 1a,b) [65,66,67,68,69]. However, the intramolecular version of the catalytic oxidative cycloaddition of aldoximes is unknown so far. In the present work, we have developed an efficient synthesis of fused isoxazole derivatives using this approach.

## 2. Results and Discussion

In order to find the optimal conditions for intramolecular cycloaddition, alkyne-tethered aldoxime **1a** was treated with a catalytic amount of iodine reagent **2**, *p*-toluenesulfonic acid and *m*-CPBA in various solvents at room temperature (Table 1). After the screening of solvents for this reaction (entries 1–8), dichloromethane was found to be the best solvent and the target compound **3a** was obtained in a 94% yield (entry 1). However, decreasing the amount of *p*-toluenesulfonic acid or using trifluoromethanesulfonic acid instead of *p*-toluenesulfonic acid resulted in lower yields of the desired product **3a** (entries 9–11). These results indicated that the addition of *p*-toluenesulfonic acid was necessary for the intramolecular cycloaddition of aldoxime **1a**. In addition, when the reaction time was shortened, the yield of the desired product **3a** was decreased (entry 12). Moreover, we observed a decline of the yield of the target product when 5 mol% and 1 mol% of 2-iodobenzoic acid **2a** were used (entries 13–14). Thus, 10 mol% of 2-iodobenzoic acid **2a** is the most suitable for the reaction. Other iodine reagents **2** were found less efficient (entries 1, 15–18).

Having in hand optimal reaction conditions, we performed the catalytic intramolecular cycloaddition of various alkyne- or alkene-tethered aldoximes **1** under optimized conditions (Figure 2). It should be pointed out that all starting compounds can be prepared very efficiently from the corresponding salicylaldehydes. It was found that the reaction is very general both for alkene and acetylene-derived starting materials to form the corresponding condensed heterocycles **3a–j**. The structure of product **3c** was established by X-ray crystallography. The intramolecular cycloaddition of aldoximes **1a–j** containing electron-donating or electron-withdrawing groups in the molecule afforded the desired products **3a–j** in up to a 91% yield. Furthermore, this catalytic system was also effective in the reaction of alkene-tethered aldoximes **1k–s**, and the desired isoxazoline-fused cyclic products **3k–s** were obtained in up to a 90% yield. In comparison with other approaches [37,40,70] to the synthesis of fused isoxazoles and isoxazolines, our method is robust, affords comparable or higher yields of desired products, is easy to perfrom and does not require the use of excess oxidant or heating for the generation of intermediate–nitrile oxides. In addition, especially interesting is the possibility to perform the reaction with internal alkyne **1t** or alkenes **1u,v**. The respective products **3t–v** were isolated in 40–90% yields.

To explore the mechanism of this reaction, several control experiments have been performed (Figure 3, and see the Appendix A for details: Appendix A). The key point of the reaction is the generation of the active hypervalent iodine species, which mediates an intermediate formation. The treatment of **2a** and *m*-CPBA in the presence of *p*-toluenesulfonic acid produced hydroxy(aryl)iodonium tosylate [71], the formation of which was confirmed by ESI mass spectrometry and ^1^H NMR spectroscopy (see Appendix A for details: Appendix A). Although the similar hydroxy(aryl)iodonium species is instantaneously formed from *m*-CPBA and **2a** in the absence of *p*-toluenesulfonic acid, this species is immediately converted to 2-iodosylbenzoic acid (IBA **4**), which cannot be applied for the intramolecular cycloaddition of aldoxime **1a** (Table 1, entry 10 and Figure 3, reaction (a)). Therefore, it was expected that *p*-toluenesulfonic acid would play a very significant role in the generation and supply of the active species. Actually, the reaction of **1a** with **4** in the presence of a catalytic amount of *p*-toluenesulfonic acid produced the desired compound **3a** in a 79% yield (reaction (b)). At the same time, we suggested that the active species can be formed with the 3-chlorobenzoic acid, which is produced during the oxidation of 2-iodobenzoic acid by *m*-CPBA. The addition of 3-chlorobenzoic acid instead of *p*-toluenesulfonic acid has not yielded **3a**, and **1a** was recovered from the reaction mixture (reaction (c)). These results indicate that the presence of a catalytic amount of *p*-toluenesulfonic acid in this reaction is sufficient to work in the reaction systems as well as contribute significantly to the formation of the active species. The reaction proceeds only in the case of the stronger acid *p*-TsOH (p*K*_a_ = −2.8), but not 3-chlorobenzoic acid (p*K*_a_ = 3.8). Additionally, we have found that the reaction of protected oxime **5** under optimized conditions did not yield the desired product **3a** (reaction (d)), and the starting compound **5** was recovered from the reaction mixture. This experiment confirms a ligand exchange of hypervalent iodine species with aldoxime and subsequent nitrile oxide formation [62].

Based on these control experiments and the related reactions of hypervalent iodine(III) compounds [37,59,69,70,72,73], we proposed the reaction mechanism (Figure 4). Hydroxy(aryl)iodonium tosylate **6** plays the role of the active species. It is produced by the reaction of *p*-toluenesulfonic acid with **4**, which is generated from *m*-CPBA and **2a**. The intermediate **6** reacts with aldoxime **1** via the ligand exchange reaction to produce iodonium intermediate **7**. Next, nitrile oxide **8** is formed by the elimination of **2a** and *p*-toluenesulfonic acid. Subsequent intramolecular cycloaddition results in the desired isoxazole derivatives **3**. Finally, the regenerated **2a** reacts with *m*-CPBA to continue the next catalytic reaction cycle.

## 3. Materials and Methods

### 3.1. General Experimental Remarks

All commercial reagents were ACS grade reagents and used without further purification from freshly opened containers. All solvents were distilled prior to use. Melting points were determined in an open capillary tube with Buchi M-580 melting point apparatus. Infrared spectra were recorded as ATR on a P Agilent Cary 630 FT-IR spectrophotometer. NMR spectra were recorded on a Bruker BioSpin NMR spectrometer at 400 or 600 MHz ((^1^H NMR), 101 or 150 MHz (^13^C NMR), 376 MHz (^19^F NMR)). Chemical shifts are reported in parts per million (ppm). High-resolution mass spectrometry measurements were performed using a Shimadzu LCMS-9030 Q-TOF mass spectrometer, coupled with LC-30 UHPLC system. X-ray crystal analysis was performed by Rigaku XtaLAB Synergy, single source at home/near, HyPix using CuKα radiation (λ = 1.54184 Å) at 105 K. Please see the Appendix A or the cif file for more detailed crystallography information. The (*E*)-2-(Prop-2-yn-1-yloxy)benzaldehyde *O*-methyl oxime **5** was prepared according to the reported procedure [74].

### 3.2. General Cyclization Procedure of 2-Alkoxyaldoximes 1

The 2-Iodobenzoic acid **2a** (5.0 mg, 0.020 mmol), *m*-CPBA (52 mg, 0.30 mmol) and *p*-TsOH•H_2_O (7.6 mg, 0.040 mmol) were added to 2-alkoxybenzaldehyde oximes **1** (0.20 mmol) in dichloromethane (2 mL). The reaction mixture was stirred at room temperature for 24 h. After the completion reaction, saturated NaHCO_3_ (15 mL), water (5 mL) and then dichloromethane (3 mL) were added, and the mixture was extracted with dichloromethane. The organic layer was dried with MgSO_4_ and concentrated under reduced pressure. Purification by column chromatography (hexane-CH_2_Cl_2_ = 3:1) afforded the pure product **3**.

4*H*-Chromeno [4,3-*c*]isoxazole (**3a**) [37]: Reaction of (*E*)-2-(prop-2-yn-1-yloxy)benzaldehyde oxime **1a** (34 mg, 0.20 mmol) according to the general procedure afforded 32 mg (94%) of product **3a**, isolated as yellowish oil; IR (ATR) cm^−1^: 3118, 3059, 2921, 2866, 1614, 1470, 1360, 1213, 1109, 765, 743; ^1^H NMR (400 MHz, CDCl_3_): δ 8.21 (t, *J* = 1.2 Hz, 1H), 7.88 (dd, *J* = 7.6, 1.6 Hz, 1H), 7.40–7.32 (m, 1H), 7.10–7.05 (m, 1H), 7.02 (dd, *J* = 8.2, 1.0 Hz, 1H), 5.24 (d, *J* = 1.2 Hz, 2H); ^13^C NMR (101 MHz, CDCl_3_): δ 155.0, 153.8, 150.8, 132.3, 124.7, 122.6, 118.0, 114.1, 111.3, 61.5; HRMS (ESI-positive mode): calcd for C_10_H_8_NO_2_ [M + H]^+^, 174.0550, found, 174.0550.

Large scale reaction for preparation of 4*H*-Chromeno [4,3-*c*]isoxazole (**3a**) [37]: Reaction of (*E*)-2-(prop-2-yn-1-yloxy)benzaldehyde oxime **1a** (1000 mg, 5.71 mmol) according to the general procedure afforded 951 mg (96%) of product **3a**, isolated as yellowish oil.

*8-Fluoro-4H-chromeno [4,3-c]isoxazole (**3b**)*: Reaction of (*E*)-5-fluoro-2-(prop-2-yn-1-yloxy)benzaldehyde oxime **1b** (38 mg, 0.20 mmol) according to the general procedure afforded 32 mg (84%) of product **3b**, isolated as colorless solid: mp 103.0–104.2 °C; IR (ATR) cm^−1^: 3126, 3074, 2933, 1624, 1478, 1243, 1172, 1107, 783, 740; ^1^H NMR (400 MHz, CDCl_3_): δ 8.24 (t, *J* = 1.1 Hz, 1H), 7.56 (dd, *J* = 8.0, 2.8 Hz, 1H), 7.11–7.03 (m, 1H), 6.99 (dd, *J* = 9.0, 4.6 Hz, 1H), 5.23 (d, *J* = 1.1 Hz, 2H); ^13^C NMR (101 MHz, CDCl_3_): δ 157.9 (d, ^1^*J_CF_* = 242.3 Hz), 153.6 (d, ^4^*J_CF_* = 2.5 Hz), 151.1, 151.1 (d, ^4^*J_CF_* = 2.4 Hz), 119.4 (d, ^3^*J_CF_* = 8.1 Hz), 119.2 (d, ^2^*J_CF_* = 23.8 Hz), 114.9 (d, ^3^*J_CF_* = 9.2 Hz), 111.3, 110.9 (d, ^2^*J_CF_* = 24.8 Hz), 61.6; ^19^F NMR (376 MHz, CDCl_3_): δ -120.1; HRMS (ESI-positive mode): calcd for C_10_H_7_FNO_2_ [M + H]^+^, 192.0455; found, 192.0456.

*8-Chloro-4H-chromeno [4,3-c]isoxazole (**3c**)* [75]: Reaction of (*E*)-5-chloro-2-(prop-2-yn-1-yloxy)benzaldehyde oxime **1c** (41 mg, 0.20 mmol) according to the general procedure afforded 29 mg (71%) of product **3c**, isolated as colorless solid: mp 127.1–128.7 °C (lit. [75]; 122.0 °C); IR (ATR) cm^−1^: 3116, 3072, 2920, 1611, 1469, 1355, 1212, 1127, 1083, 766, 742; ^1^H NMR (400 MHz, CDCl_3_): δ 8.24 (t, *J* = 1.2 Hz, 1H), 7.85 (d, *J* = 2.8 Hz, 1H), 7.30 (dd, *J* = 8.8, 2.8 Hz, 1H), 6.97 (d, *J* = 8.8 Hz, 1H), 5.25 (d, *J* = 1.2 Hz, 2H).; ^13^C NMR (101 MHz, CDCl_3_): δ 153.4, 153.1, 151.2, 132.1, 127.6, 124.4, 119.5, 115.3, 111.1, 61.7; HRMS (ESI-positive mode): calcd for C_10_H_7_^35^ClNO_2_ [M + H]^+^, 208.0160; found, 208.0160.

*8-Bromo-4H-chromeno [4,3-c]isoxazole (**3d**)* [76]: Reaction of (*E*)-5-bromo-2-(prop-2-yn-1-yloxy)benzaldehyde oxime **1d** (50 mg, 0.20 mmol) according to the general procedure afforded 37 mg (74%) of product **3d**, isolated as yellowish solid: mp 120.2–120.9 °C (lit. [76], 118.0–119.0 °C); IR (ATR) cm^−1^: 3112, 3067, 2922, 1607, 1464, 1353, 1210, 1128, 758; ^1^H NMR (400 MHz, CDCl_3_): δ 8.24 (t, *J* = 1.0 Hz, 1H), 8.00 (d, *J* = 2.6 Hz, 1H), 7.44 (dd, *J* = 8.8, 2.6 Hz, 1H), 6.91 (d, *J* = 8.8 Hz, 1H), 5.25 (d, *J* = 1.0 Hz, 2H).; ^13^C NMR (101 MHz, CDCl_3_): δ 153.9, 152.9, 151.2, 135.0, 127.3, 119.9, 115.7, 114.8, 111.0, 61.6; HRMS (ESI-positive mode): calcd for C_10_H_7_^79^BrNO_2_ [M + H]^+^, 251.9655; found, 251.9650.

*6,8-Dibromo-4H-chromeno [4,3-c]isoxazole (**3e**)*: Reaction of (*E*)-3,5-dibromo-2-(prop-2-yn-1-yloxy)benzaldehyde oxime **1e** (66 mg, 0.20 mmol) according to the general procedure afforded 53 mg (80%) of product **3e**, isolated as yellowish solid: mp 149.0–150.0 °C; IR (ATR) cm^−1^: 3138, 3123, 3065, 2919, 1597, 1450, 1375, 1217, 1117, 787, 749.; ^1^H NMR (400 MHz, CDCl_3_): δ 8.28 (t, *J* = 1.2 Hz, 1H), 7.97 (d, *J* = 2.2 Hz, 1H), 7.73 (d, *J* = 2.2 Hz, 1H), 5.37 (d, *J* = 1.2 Hz, 2H).; ^13^C NMR (101 MHz, CDCl_3_): δ 152.6, 151.6, 150.8, 137.7, 126.5, 116.6, 114.8, 113.0, 110.8, 62.5.; HRMS (ESI-positive mode): calcd for C_10_H_6_^79^Br_2_NO_2_ [M + H]^+^, 329.8760; found, 329.8755.

*8-Nitro-4H-chromeno [4,3-c]isoxazole (**3f**)*: Reaction of (*E*)-5-nitro-2-(prop-2-yn-1-yloxy)benzaldehyde oxime **1f** (44 mg, 0.20 mmol) according to the general procedure afforded 40 mg (91%) of product **3f**, isolated as colorless solid: mp 221.7–222.1 °C; IR (ATR) cm^−1^: 3090, 2948, 1620, 1582, 1528, 1507, 1475, 1340, 1226, 1129, 749.; ^1^H NMR (400 MHz, CDCl_3_): δ 8.81 (d, *J* = 2.6 Hz, 1H), 8.41–8.27 (m, 1H), 8.24 (dd, *J* = 9.2, 2.6 Hz, 1H), 7.13 (d, *J* = 9.2 Hz, 1H), 5.43 (d, *J* = 1.2 Hz, 1H).; ^13^C NMR (101 MHz, CDCl_3_): δ 159.5, 152.2, 151.9, 142.6, 127.5, 120.9, 118.8, 113.9, 110.2, 62.6.; HRMS (ESI-positive mode): calcd for C_10_H_7_N_2_O_4_ [M + H]^+^, 219.0400; found, 219.0399.

*8-Methyl-4H-chromeno [4,3-c]isoxazole (**3g**)* [39]: Reaction of (*E*)-5-methyl-2-(prop-2-yn-1-yloxy)benzaldehyde oxime **1g** (37 mg, 0.20 mmol) according to the general procedure afforded 30 mg (81%) of product **3g**, isolated as yellowish solid: mp 103.6–105.2 °C (lit. [39]; 103.0–104.0 °C); IR (ATR) cm^−1^: 3101, 3060, 2920, 1620, 1577, 1487, 1460, 1359, 1212, 1130, 783, 745; ^1^H NMR (400 MHz, CDCl_3_): δ 8.19 (t, *J* = 1.2 Hz, 1H), 7.69 (d, *J* = 2.0 Hz, 1H), 7.16 (dd, *J* = 8.2, 2.0 Hz, 1H), 6.92 (d, *J* = 8.2 Hz, 1H), 5.21 (d, *J* = 1.2 Hz, 2H), 2.34 (s, 3H); ^13^C NMR (101 MHz, CDCl_3_): δ 154.0, 152.9, 150.7, 133.0, 132.1, 124.8, 117.7, 113.8, 111.5, 61.4, 20.8.; HRMS (ESI-positive mode): calcd for C_11_H_10_NO_2_ [M + H]^+^, 188.0706; found, 188.0708.

*4H-Benzo* [5,6]*chromeno [4,3-c]isoxazole (**3h**)* [77]: Reaction of (*E*)-2-(prop-2-yn-1-yloxy)-1-naphthaldehyde oxime **1h** (45 mg, 0.20 mmol) according to the general procedure afforded 30 mg (67%) of product **3h**, isolated as yellowish solid: mp 175.0–176.0 °C (lit. [77]; 180.0–181.0 °C); IR (ATR) cm^−1^: 3107, 2924, 2870, 1591, 1519, 1443, 1357, 1221, 1119, 770, 748; ^1^H NMR (400 MHz, CDCl_3_): δ 9.05 (d, *J* = 8.0 Hz, 1H), 8.27 (t, *J* = 1.2 Hz, 1H), 7.87–7.83 (m, 1H), 7.83–7.79 (m, 1H), 7.68–7.62 (m, 1H), 7.49–7.43 (m, 1H), 7.20 (d, *J* = 8.8 Hz, 1H), 5.33 (d, *J* = 1.2 Hz, 2H); ^13^C NMR (101 MHz, CDCl_3_): δ 154.9, 154.6, 149.7, 133.0, 130.2, 129.8, 128.5, 128.5, 126.5, 125.0, 118.7, 111.9, 108.0, 61.5.; HRMS (ESI-positive mode): calcd for C_14_H_10_NO_2_ [M + H]^+^, 224.0706; found, 224.0704.

*9-Chloro-4H-chromeno [4,3-c]isoxazole (**3i**)*: Reaction of (*E*)-2-chloro-6-(prop-2-yn-1-yloxy)benzaldehyde oxime **1i** (42 mg, 0.20 mmol) according to the general procedure afforded 37 mg (88%) of product **3i**, isolated as colorless solid: mp 100.5–101.0 °C; IR (ATR) cm^−1^: 3100, 2926, 1600, 1454, 1406, 1364, 1219, 1151, 1099, 780, 742; ^1^H NMR (400 MHz, CDCl_3_): δ 8.30 (t, *J* = 1.2 Hz, 1H), 7.30–7.24 (m, 1H), 7.17 (dd, *J* = 8.0, 1.2 Hz, 1H), 6.98 (dd, *J* = 8.0, 1.2 Hz, 1H), 5.21 (d, *J* = 1.2 Hz, 2H).; ^13^C NMR (101 MHz, CDCl_3_): δ 156.4, 152.9, 150.6, 132.5, 131.8, 124.6, 116.7, 114.4, 112.2, 61.3; HRMS (ESI-positive mode): calcd for C_10_H_7_^35^ClNO_2_ [M + H]^+^, 208.0160; found, 208.0164.

*6-Chloro-4H-chromeno [4,3-c]isoxazole (**3j**)*: Reaction of (*E*)-3-chloro-6-(prop-2-yn-1-yloxy)benzaldehyde oxime **1j** (42 mg, 0.20 mmol) according to the general procedure afforded 36 mg (86%) of product **3j**, isolated as colorless solid: mp 112.3–113.3 °C; IR (ATR) cm^−1^: 3119, 2923, 1605, 1467, 1433, 1355, 1222, 1145, 1085, 786, 734; ^1^H NMR (400 MHz, CDCl_3_): δ 8.27 (t, *J* = 1.2 Hz, 1H), 7.80 (dd, *J* = 8.0, 1.6 Hz, 1H), 7.44 (dd, *J* = 8.0, 1.6 Hz, 1H), 7.02 (t, *J* = 8.0 Hz, 1H), 5.37 (d, *J* = 1.2 Hz, 2H).; ^13^C NMR (101 MHz, CDCl_3_): δ 153.4, 151.2, 150.7, 132.7, 123.1, 122.8, 115.6, 111.0, 62.3; HRMS (ESI-positive mode): calcd for C_10_H_7_^35^ClNO_2_ [M + H]^+^, 208.0160; found, 208.0161.

*3a,4-Dihydro-3H-chromeno [4,3-c]isoxazole (**3k**)* [37]: Reaction of (*E*)-2-(allyloxy)benzaldehyde oxime **1k** (35 mg, 0.20 mmol) according to the general procedure afforded 28 mg (80%) of product **3k**, isolated as pale solid: mp 59.7–61.0 °C (lit. [37] 60–61 °C); IR (ATR) cm^−1^: 3075, 2995, 2933, 2880, 1607, 1467, 1458, 1228, 1155, 760; ^1^H NMR (400 MHz, CDCl_3_): δ 7.79 (dd, *J* = 8.0, 1.6 Hz, 1H), 7.39–7.30 (m, 1H), 7.03–6.97 (m, 2H), 6.95 (dd, *J* = 8.4, 1.2 Hz, 1H), 4.75–4.64 (m, 2H), 4.14–4.04 (m, 1H), 4.01–3.86 (m, 2H); ^13^C NMR (101 MHz, CDCl_3_): δ 155.7, 152.9, 132.6, 125.8, 122.0, 117.5, 113.1, 70.7, 69.4, 46.0; HRMS (ESI-positive mode): calcd for C_10_H_10_NO_2_ [M + H]^+^, 176.0706; found, 176.0710.

*8-Fluoro-3a,4-dihydro-3H-chromeno [4,3-c]isoxazole (**3l**)*: Reaction of (*E*)-2-(allyloxy)-5-fluorobenzaldehyde oxime **1l** (39 mg, 0.20 mmol) according to the general procedure afforded 33 mg (85%) of product **3l**, isolated as colorless solid: mp 145.3–146.4 °C; IR (ATR) cm^−1^: 3063, 2932, 2884, 1614, 1481, 1459, 1301, 1235, 1171, 1121, 741; ^1^H NMR (400 MHz, CDCl_3_): δ 7.44 (dd, *J* = 8.2, 3.0 Hz, 1H), 7.08–7.00 (m, 1H), 6.91 (dd, *J* = 9.0, 4.6 Hz, 1H), 4.74–4.63 (m, 2H), 4.10–4.01 (m, 1H), 3.98–3.85 (m, 2H); ^13^C NMR (101 MHz, CDCl_3_): δ 157.3 (d, *^1^J_CF_* = 242.1 Hz), 152.5 (d, *^4^J_CF_* = 2.4 Hz), 151.9 (d, *^4^J_CF_* = 1.9 Hz), 119.9 (d, *^2^J_CF_* = 24.1 Hz), 118.9 (d, *^3^J_CF_* = 8.1 Hz), 113.7 (d, *^3^J_CF_* = 8.8 Hz), 111.2 (d, *^2^J_CF_* = 24.4 Hz), 71.0, 69.4, 45.6; ^19^F NMR (376 MHz, CDCl_3_) δ -121.0.; HRMS (ESI-positive mode): calcd for C_10_H_9_FNO_2_ [M + H]^+^, 194.0612; found, 194.0605.

*8-Chloro-3a,4-dihydro-3H-chromeno [4,3-c]isoxazole (**3m**)* [37]: Reaction of (*E*)-2-(allyloxy)-5-chlorobenzaldehyde oxime **1m** (42 mg, 0.20 mmol) according to the general procedure afforded 34 mg (81%) of product **3m**, isolated as colorless solid: mp 129.7–130.1 °C (lit. [37] 129–130 °C); IR (ATR) cm^−1^: 3038, 2923, 2873, 1610, 1474, 1443, 1226, 1132, 734; ^1^H NMR (400 MHz, CDCl_3_): δ 7.76 (d, *J* = 2.6 Hz, 1H), 7.28 (dd, *J* = 9.1, 2.6 Hz, 1H), 6.90 (d, *J* = 9.1 Hz, 1H), 4.75–4.65 (m, 2H), 4.11–4.02 (m, 1H), 3.98–3.85 (m, 2H); ^13^C NMR (101 MHz, CDCl_3_): δ 154.1, 152.0, 132.4, 127.0, 125.1, 119.0, 114.3, 71.0, 69.4, 45.5; HRMS (ESI-positive mode): calcd for C_10_H_9_^35^ClNO_2_ [M + H]^+^, 210.0316; found, 210.0319.

*8-Bromo-3a,4-dihydro-3H-chromeno [4,3-c]isoxazole (**3n**)* [37]: Reaction of (*E*)-2-(allyloxy)-5-bromobenzaldehyde oxime **1n** (51 mg, 0.20 mmol) according to the general procedure afforded 41 mg (80%) of product **3n**, isolated as colorless solid: mp 126.7–127.8 °C (lit. [37] 125.0–127 °C); IR (ATR) cm^−1^: 2985, 2924, 2870, 1603, 1474, 1436, 1206, 1131, 728; ^1^H NMR (400 MHz, CDCl_3_): δ 7.92 (d, *J* = 2.4 Hz, 1H), 7.41 (dd, *J* = 8.8, 2.4 Hz, 1H), 6.85 (d, *J* = 8.8 Hz, 1H), 4.77–4.66 (m, 2H), 4.11–4.03 (m, 1H), 3.98–3.85 (m, 2H).; ^13^C NMR (101 MHz, CDCl_3_): δ 154.6, 151.9, 135.3, 128.2, 119.4, 114.9, 114.3, 71.0, 69.4, 45.5; HRMS (ESI-positive mode): calcd for C_10_H_9_^79^BrNO_2_ [M + H]^+^, 253.9811; found, 253.9812.

*8-Nitro-3a,4-dihydro-3H-chromeno [4,3-c]isoxazole (**3o**)* [37]: Reaction of (*E*)-2-(allyloxy)-5-nitrobenzaldehyde oxime **1o** (44 mg, 0.20 mmol) according to the general procedure afforded 35 mg (80%) of product **3o**, isolated as colorless solid: 220.0–221.0 °C (lit. [37] 215.0–217.0 °C); IR (ATR) cm^−1^: 3071, 3022, 2923, 1608, 1576, 1513, 1454, 1342, 1232, 1127, 839, 745; ^1^H NMR (400 MHz, CDCl_3_): δ 8.72 (d, *J* = 2.8 Hz, 1H), 8.21 (dd, *J* = 9.2, 2.8 Hz, 1H), 7.08 (d, *J* = 9.2 Hz, 1H), 4.87–4.76 (m, 2H), 4.22–4.14 (m, 1H), 4.05–3.94 (m, 2H).; ^13^C NMR (101 MHz, CDCl_3_): δ 159.8, 151.1, 142.4, 127.4, 122.1, 118.5, 113.5, 71.3, 70.0, 45.0.; HRMS (ESI-positive mode): calcd for C_10_H_9_N_2_O_4_ [M + H]^+^, 221.0557; found, 221.0553.

*8-Methyl-3a,4-dihydro-3H-chromeno [4,3-c]isoxazole (**3p**)* [70]: Reaction of (*E*)-2-(allyloxy)-5-methylbenzaldehyde oxime **1p** (38 mg, 0.20 mmol) according to the general procedure afforded 31 mg (82%) of product **3p**, isolated as yellowish solid: 140.7–141.8 °C (lit. [70] 142 °C); IR (ATR) cm^−1^: 3058, 2995, 2915, 2875, 1606, 1484, 1229, 1133, 744; ^1^H NMR (400 MHz, CDCl_3_): δ 7.60 (d, *J* = 2.2 Hz, 1H), 7.14 (dd, *J* = 8.4, 2.4 Hz, 1H), 6.85 (d, *J* = 8.4 Hz, 1H), 4.76–4.62 (m, 2H), 4.11–4.01 (m, 1H), 3.99–3.84 (m, 2H), 2.30 (s, 3H).; ^13^C NMR (101 MHz, CDCl_3_): δ 153.7, 153.1, 133.6, 131.4, 125.6, 117.3, 112.7, 70.7, 69.4, 46.1, 20.6; HRMS (ESI-positive mode): calcd for C_11_H_12_NO_2_ [M + H]^+^, 190.0863; found, 190.0869.

*3a,4-Dihydro-3H-benzo* [5,6]*chromeno [4,3-c]isoxazole (**3q**)* [37]: Reaction of (*E*)-2-(allyloxy)-1-naphthaldehyde oxime **1q** (45 mg, 0.20 mmol) according to the general procedure afforded 28 mg (62%) of product **3q**, isolated as yellowish solid: 75.0–75.8 °C (lit. [37] 78–80 °C); IR (ATR) cm^−1^: 3052, 2999, 2935, 2879, 1621, 1578, 1512, 1440, 1227, 1123, 747; ^1^H NMR (400 MHz, CDCl_3_): δ 9.03 (dd, *J* = 8.8, 1.0 Hz, 1H), 7.85–7.75 (m, 2H), 7.65–7.57 (m, 1H), 7.48–7.39 (m, 1H), 7.11 (d, *J* = 9.2 Hz, 1H), 4.82–4.75 (m, 1H), 4.75–4.68 (m, 1H), 4.25 (dd, *J* = 8.8, 1.0 Hz, 1H), 4.17–4.05 (m, 1H), 3.94 (dd, *J* = 12.7, 8.0 Hz, 1H).; ^13^C NMR (101 MHz, CDCl_3_): δ 155.8, 153.3, 133.6, 130.6, 129.4, 128.7, 128.5, 126.7, 124.9, 118.3, 106.2, 69.6, 69.3, 47.1; HRMS (ESI-positive mode): calcd for C_14_H_12_NO_2_ [M + H]^+^, 226.0863; found, 226.0862.

*9-Chloro-3a,4-dihydro-3H-chromeno [4,3-c]isoxazole (**3r**)*: Reaction of 2-(allyloxy)-6-chlorobenzaldehyde **1r** (42 mg, 0.20 mmol) according to the general procedure afforded 35 mg (83%) of product **3r**, isolated as colorless solid: 145.2–145.8 °C; IR (ATR) cm^−1^: 3091, 2986, 2932, 2871, 1588, 1482, 1442, 1228, 1178, 1149, 726; ^1^H NMR (400 MHz, CDCl_3_): δ 7.25–7.19 (m, 1H), 7.08 (dd, *J* = 7.8, 1.1 Hz, 1H), 6.88 (dd, *J* = 8.4, 1.1 Hz, 1H), 4.72–4.64 (m, 2H), 4.14–4.04 (m, 1H), 4.01–3.88 (m, 2H); ^13^C NMR (101 MHz, CDCl_3_): δ 156.7, 151.3, 133.2, 131.8, 123.9, 116.1, 112.7, 69.9, 69.0, 46.4; HRMS (ESI-positive mode): calcd for C_10_H_9_^35^ClNO_2_ [M + H]^+^, 210.0316; found, 210.0320.

*6-Chloro-3a,4-dihydro-3H-chromeno [4,3-c]isoxazole (**3s**)*: Reaction of 2-(allyloxy)-3-chlorobenzaldehyde **1s** (42 mg, 0.20 mmol) according to the general procedure afforded 36 mg (86%) of product **3s**, isolated as colorless solid: mp 104.7–105.5 °C; IR (ATR) cm^−1^: 3073, 2988, 2929, 2867, 1600, 1468, 1438, 1230, 1145, 1079, 727; ^1^H NMR (400 MHz, CDCl_3_): δ 7.72 (dd, *J* = 8.0, 1.6 Hz, 1H), 7.42 (dd, *J* = 7.6, 1.6 Hz, 1H), 6.98–6.92 (m, 1H), 4.89–4.81 (m, 1H), 4.81–4.69 (m, 1H), 4.23–4.10 (m, 1H), 4.04–3.90 (m, 2H).; ^13^C NMR (101 MHz, CDCl_3_): δ 152.3, 151.3, 132.8, 124.3, 122.6, 122.2, 114.8, 71.0, 70.0, 45.6; HRMS (ESI-positive mode): calcd for C_10_H_9_^35^ClNO_2_ [M + H]^+^, 210.0316; found, 210.0313.

*3-Phenyl-4H-chromeno [4,3-c]isoxazole (**3t**)*: Reaction of (*E*)-2-((3-phenylprop-2-yn-1-yl)oxy)benzaldehyde oxime **1t** (50 mg, 0.20 mmol) according to the general procedure afforded 45 mg (90%) of product **3t**, isolated as colorless solid: mp 156.0–157.0 °C; IR (ATR) cm^−1^: 3059, 2924, 2875, 1612, 1577, 1474, 1446, 1420, 1375, 1299, 1221, 1101, 756, 744; ^1^H NMR (400 MHz, CDCl_3_) δ 7.89 (dd, *J* = 7.8, 1.6 Hz, 1H), 7.65–7.59 (m, 2H), 7.54–7.43 (m, 3H), 7.39–7.33 (m, 1H), 7.11–7.05 (m, 1H), 7.03 (dd, *J* = 8.2, 1.0 Hz, 1H), 5.45 (s, 2H).; ^13^C NMR (101 MHz, CDCl_3_) δ 161.7, 155.1, 154.7, 132.2, 130.3, 129.3, 127.4, 126.3, 124.5, 122.4, 117.8, 114.2, 106.7, 62.6; HRMS (ESI-positive mode): calcd for C_16_H_12_NO_2_ [M + H]^+^, 250.0863; found, 250.0863.

*3-Phenyl-3a,4-dihydro-3H-chromeno [4,3-c]isoxazole (**3u**)* [37]: Reaction of (*E*)-2-(cinnamyloxy)benzaldehyde oxime **1u** (51 mg, 0.20 mmol) according to the general procedure afforded 20 mg (40%) of product **3u**, isolated as colorless solid: mp 151.9–152.9 °C (lit. [37] 156–158 °C); IR (ATR) cm^−1^: 3037, 2999, 2927, 2884, 1600, 1467, 1454, 1233, 1218, 1119, 1032, 999, 754; ^1^H NMR (400 MHz, CDCl_3_): δ 7.77 (dd, *J* = 7.6, 1.6 Hz, 1H), 7.42–7.31 (m, 5H), 7.30–7.24 (m, 1H), 6.96 (t, *J* = 7.6 Hz, 1H), 6.88 (d, *J* = 8.4 Hz, 1H), 5.18 (d, *J* = 12.6 Hz, 1H), 4.58 (dd, *J* = 10.4, 2.0 Hz, 1H), 4.17 (dd, *J* = 12.4, 10.4 Hz, 1H), 3.89–3.78 (m, 1H). ^13^C NMR (101 MHz, CDCl_3_): δ 154.5, 152.3, 136.2, 131.6, 127.9, 125.6, 124.5, 120.9, 116.4, 112.1, 84.7, 68.0, 51.9; HRMS (ESI-positive mode): calcd for C_16_H_14_NO_2_ [M + H]^+^, 252.1019; found, 252.1020.

*2a,2a^1^,3,4,5,5a-Hexahydroxantheno [9,1-cd]isoxazole (**3v**)* [37]: Reaction of (*E*)-2-(cyclohex-2-en-1-yloxy)benzaldehyde oxime **1v** (42 mg, 0.20 mmol) according to the general procedure afforded 30 mg (70%) of product **3v**, isolated as colorless solid: mp 104.7–105.5 °C (lit. [37] 103–104 °C); IR (ATR) cm^−1^: 2492, 2924, 2862, 1600. 1573, 1493,1458, 1380, 1344, 1319, 1292, 1264, 1227, 1207, 1158, 1113, 1029, 999, 901, 868, 840, 812, 754, 710, 649, 516, 450; ^1^H NMR (600 MHz, CDCl_3_) δ 7.86 (dd, *J* = 7.8, 1.8 Hz, 1H), 7.37–7.30 (m, 1H), 7.00–6.96 (m, 1H), 6.94 (dd, *J* = 8.1, 0.9 Hz, 1H), 4.93 (m, 1H), 4.74 (m, 1H), 3.82 (m, 1H), 2.06–1.96 (m, 2H), 1.66–1.59 (m, 1H), 1.44–1.35 (m, 1H), 1.35–1.24 (m, 1H), 1.11–1.01 (m, 1H); ^13^C NMR (151 MHz, CDCl_3_) δ 153.9, 153.6, 132.8, 125.4, 121.5, 118.1, 112.8, 80.3, 74.8, 47.4, 27.8, 27.2, 17.3; HRMS (ESI-positive mode): calcd for C_13_H_13_NO_2_ [M + H]^+^ calcd for C_13_H_14_NO_2_^+^, 216.1025; found, 216.1021.

## 4. Conclusions

We have developed a reliable and efficient method for the synthesis of diverse fused isoxazoles and isoxazolines via catalytic intramolecular oxidative cycloaddition of aldoximes with the use of hypervalent iodine species. The reaction mechanism was studied in detail by various spectroscopic methods and control experiments. It was found that the key intermediate is hydroxy(aryl)iodonium tosylate. This hypervalent iodine derivative is generated in situ from 2-iodobenzoic acid and *m*-CPBA in the presence of *p*-toluenesulfonic acid.

## Figures and Tables

**Figure 1 molecules-27-03860-f001:**
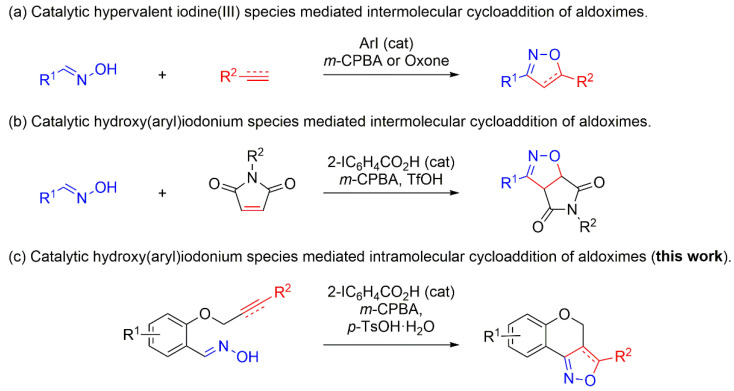
Reactions of aldoximes using catalytic hypervalent iodine(III) species.

**Figure 2 molecules-27-03860-f002:**
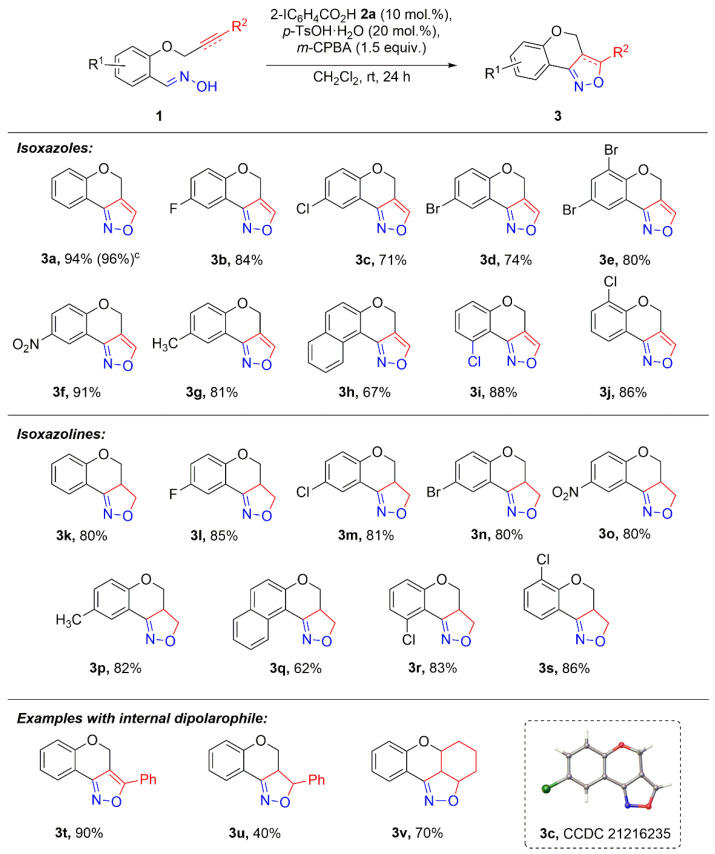
Catalytic intramolecular cycloaddition of aldoximes **1**
^a,b^. ^a^ Reaction conditions: Aldoxime **1** (0.20 mmol, 1 equiv.), **2a** (10 mol%) and *p*-toluenesulfonic acid (20 mol%) with *m*-CPBA (0.30 mmol, 1.5 equiv.) stirred in dichloromethane (2 mL) at room temperature for 24 h. ^b^ Isolated yields of **3**. ^c^ The yield of **3a** is given for 1 g scale reaction.

**Figure 3 molecules-27-03860-f003:**
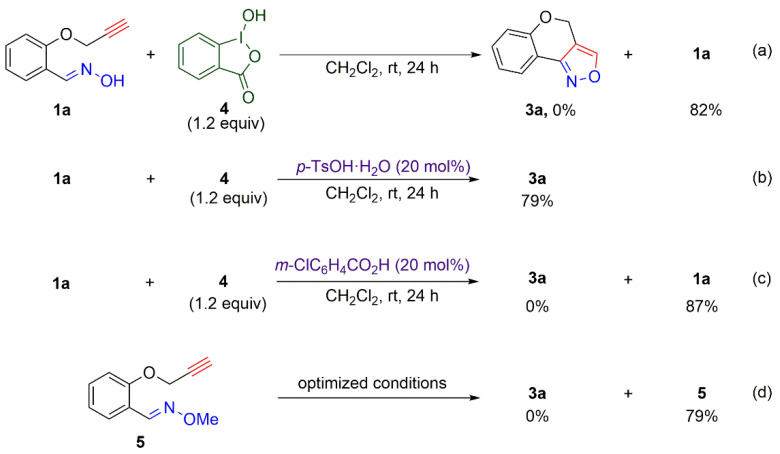
Control experiments.

**Figure 4 molecules-27-03860-f004:**
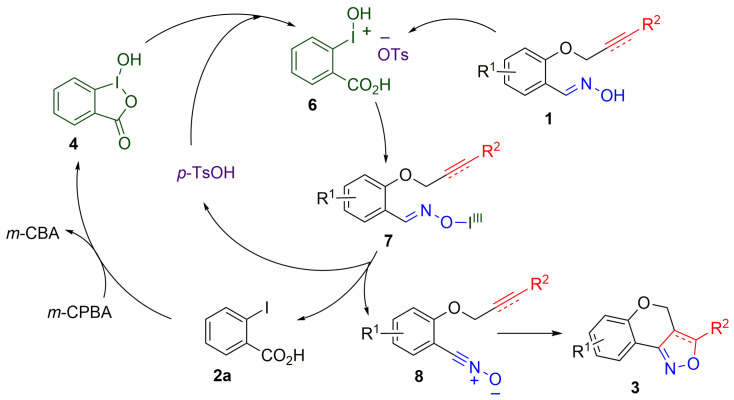
Proposed reaction mechanism.

**Table 1 molecules-27-03860-t001:** Optimization of the catalytic intramolecular cycloaddition of aldoxime **1a** ^a^.

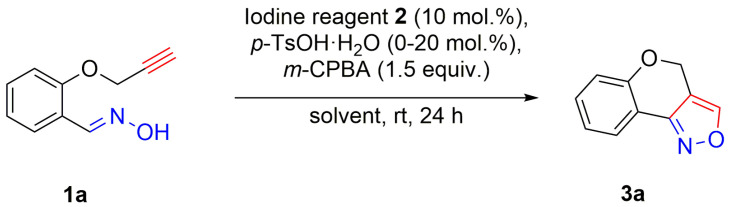				
Entry	Solvent	Iodine Reagent 2	*p*-TsOH·H_2_O (mol%)	3a Yield (%) ^b^
**1**	**CH_2_Cl_2_**	**2-IC_6_H_4_CO_2_H 2a**	**20**	**94 (94)**
2	CHCl_3_	2-IC_6_H_4_CO_2_H **2a**	20	52 (50) ^c^
3	Et_2_O	2-IC_6_H_4_CO_2_H **2a**	20	32 (31) ^c^
4	MeCN	2-IC_6_H_4_CO_2_H **2a**	20	81 (80) ^c^
5	Hexane	2-IC_6_H_4_CO_2_H **2a**	20	56 (52) ^c^
6	PhH	2-IC_6_H_4_CO_2_H **2a**	20	73 (73) ^c^
7	THF	2-IC_6_H_4_CO_2_H **2a**	20	81 (81) ^c^
8	MeOH	2-IC_6_H_4_CO_2_H **2a**	20	70 (70) ^c^
9	CH_2_Cl_2_	2-IC_6_H_4_CO_2_H **2a**	10	61 (61) ^c^
10	CH_2_Cl_2_	2-IC_6_H_4_CO_2_H **2a**	none	36 (35) ^c^
11	CH_2_Cl_2_	2-IC_6_H_4_CO_2_H **2a**	– ^d^	86 (81)
12 ^e^	CH_2_Cl_2_	2-IC_6_H_4_CO_2_H **2a**	20	73 (72) ^c^
13	CH_2_Cl_2_	2-IC_6_H_4_CO_2_H **2a** ^f^	20	81
14	CH_2_Cl_2_	2-IC_6_H_4_CO_2_H **2a** ^g^	20	62
15	CH_2_Cl_2_	PhI **2b**	20	73 (54) ^c^
16	CH_2_Cl_2_	TBAI **2c**	20	20 (20) ^c^
17	CH_2_Cl_2_	I_2_ **2d**	20	15 ^c^
18	CH_2_Cl_2_	none	20	9 ^c^

^a^ Reaction conditions: Aldoxime **1a** (0.20 mmol, 1 equiv.), iodine reagent **2** (10 mol%) and *p*-toluenesulfonic acid (0–20 mol%) with *m*-CPBA (0.30 mmol, 1.5 equiv.) stirred in solvent (2 mL) at room temperature for 12–24 h. ^b^ Yield of product **3a** determined from ^1^H NMR spectra of the reaction mixture (using as 1,2-dibromoethane as an internal standard) are shown (numbers in parentheses show an isolated yield of **3a**). ^c^ Aldoxime **1a** was detected from the reaction mixture. ^d^ TfOH was used instead of *p*-TsOH·H_2_O. ^e^ Reaction time was 12 h. ^f^ 5 mol% were used. ^g^ 1 mol% were used.

## Data Availability

Not applicable.

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
