# Peer review of "Efficient Catalytic Synthesis of Condensed Isoxazole Derivatives via Intramolecular Oxidative Cycloaddition of Aldoximes"

_molecules, 2022, doi:10.3390/molecules27123860_

Round 1
Reviewer 1 Report
I have noted that there are several determinants missing such as "Heterocycles play A key role..." or "Isoxazole fragment is among THE most popular..." as well as some repetition like "stacking INTERACTIONS and hydrophilic INTERACTIONS", other sentences that do not sound quite right and some typos like "necassary". I advise the authors to review the whole text for better readibility.
I think that exposing first the objective of this study (third paragraph) and then, the role of heterocycles and cycloaddition (1st and 2nd paragraphs) would increase the cohesion of the article, since as a reader, I don't know why I should care about heterocycles until I read the 3rd paragraph. Or even better, to begin with a detailed explanation of what is isoxazole and why is it important to research on it rather than starting with heterocycles.
I cannot understand Scheme 3. Arrows point everywhere and I do not know which compounds enter the mechanism, which exit and which react together. Please, rework this scheme.
Consider moving all the IR data in section 3.2 to the Supporting Information.
I would like to see a bit more depth into the mechanism development, since per the control experiments, I do not fully comprehend how can you reach the mechanism you proposed. More detail into why the authors think that each step represented is the actual reaction happening is needed.
Author Response
Dear Mr. Tory Tang:
Thank you for thorough evaluation of our manuscript molecules- 1770717. We are submitting the revised manuscript that addresses all comments of the referees. Below you will find our response and the list of changes made in the manuscript; our response is in bold Italic.
Reviewer(s)' Comments to Author:
Referee 1
I have noted that there are several determinants missing such as "Heterocycles play A key role..." or "Isoxazole fragment is among THE most popular..." as well as some repetition like "stacking INTERACTIONS and hydrophilic INTERACTIONS", other sentences that do not sound quite right and some typos like "necassary". I advise the authors to review the whole text for better readibility.
We agree with the comment, we have fixed this.
I think that exposing first the objective of this study (third paragraph) and then, the role of heterocycles and cycloaddition (1st and 2nd paragraphs) would increase the cohesion of the article, since as a reader, I don't know why I should care about heterocycles until I read the 3rd paragraph. Or even better, to begin with a detailed explanation of what is isoxazole and why is it important to research on it rather than starting with heterocycles.
The authors are thankful the referee for the suggestion. We think that our introduction provides the most logical approach to the subject from the most general information to more specific details.
I cannot understand Scheme 3. Arrows point everywhere and I do not know which compounds enter the mechanism, which exit and which react together. Please, rework this scheme.
The authors are thankful the referee for the remark. We have fixed it.
Consider moving all the IR data in section 3.2 to the Supporting Information.
Thank you for your suggestions but we just followed the molecules Experimental Data rules, so we do not have a problem with that.
I would like to see a bit more depth into the mechanism development, since per the control experiments, I do not fully comprehend how can you reach the mechanism you proposed. More detail into why the authors think that each step represented is the actual reaction happening is needed.
The authors are thankful for the comment. Following to your advice we have added following sentances into the main text: “The key point of the reaction is generation of active hypervalent iodine species, which mediates an intermediate formation” (lines 100-102); “At the same time, we suggested that the active species can be formed with the 3-chlorobenzoic acid, which is produced during the oxidation of 2-iodobenzoic acid by m-CPBA.” (lines 112-114) and elaborated the sentence “Additionally, we have found that the reaction of protected oxime 5 under optimized con-ditions did not yield the desired product 3a (equation 4), and starting compound 5 was recovered from the reaction mixture. This experiment confirms a ligand exchange of hy-pervalent iodine species with aldoxime and subsequent nitrile oxide formation [62]” (lines 119-123)
Author Response
Dear Mr. Tory Tang:
Thank you for thorough evaluation of our manuscript molecules- 1770717. We are submitting the revised manuscript that addresses all comments of the referees. Below you will find our response and the list of changes made in the manuscript; our response is in bold Italic.
Reviewer(s)' Comments to Author:
Referee 2
Under the Results and discussion line 69 “thus 20 mol% of 2-iodobenzoic acid 2a is most suitable……” But at the foot note of Table-1 it shows 10 mol% of 2a.
We agree with the comment. We have fixed it, thank you.
The conversions in case of internal alkynes and alkenes (Table 2 3t-v) are interesting if it would be studied with a few more examples considering electronic effects.
We are thankful for the suggestion and nice comments. We will investigate this reaction using internal multiple bond compounds in the future.
Referee 3
The method seems to be very efficient with good yields and wide substrate scopes. What is the scalability of the method? (e.g. gram scale synthesis).
We are thankful for the suggestion. The method is applicable for transformation of 1g starting aldoxime 1a into 3a with the 96% yield. We added that result in the Table 2.
The proposed reaction mechanism seems to be satisfactory. (Slight elaboration is recommended for the conversion from 6 to 7 & 8).
We agree with the comment. Following to your advice we have changed the mechanism scheme in the manuscript.
The formation of key intermediate is hydroxy(aryl)iodonium tosylate is characterized by using ESI and NMR analysis. In ESI method, the formation of intermediate is clearly visible from the m/z peak observed in the spectra. But, from 1H NMR, it is not clearly understood specially in case of step (iii) and step (iv). It is recommended to elaborate.
We are thankful for the comment. The 1H NMR analysis of generation of active species was conducted for the reaction mixtures, thus the spectra look complex and, in addition, the active species are poorly soluble in CD2Cl2, which was used for the reaction and analysis.
The authors claimed that as it is the first report of the intramolecular version of the catalytic oxidative cycloaddition of aldoximes using hypervalent iodine as a catalyst. However, authors may include the following work as a reference and may compare it with their methodology.
The thermal intramolecular cycloaddition of aldoxime was reported by Meshko and co-workers.Hassner, A.; Maurya, R.; Mesko, E. Tetrahedron Lett. 1988, 29, 5313.
Thank you for the advice, we have added the comparison and corresponding references to the main text (lines 80-83) as follow: “In comparison with other approaches [37,40,70] to the synthesis of fused isoxazoles and isoxazolines our method is robust, afford comparable or higher yields of desired products, easy perfrom and does not require the use of excess of oxidant or heating for generation of intermediate - nitrile oxides.”
We look forward to hearing from you on this manuscript.
Sincerely yours,
Mekhman S. Yusubov
Professor Mekhman S. Yusubov (Corresponding Author)
The Tomsk Polytechnic University
Tomsk, Russia, 634050
E-mail: [email protected]
Reviewer 3 Report
In this work the authors developed the “reliable and efficient method for the synthesis of diverse fused isoxazoles and isooxazolines via catalytic intramolecular oxidative cycloaddition of aldoximes with the use of hypervalent iodine species”.
The authors need to address the following comments to improve the quality of the manuscript:
1. The method seems to be very efficient with good yields and wide substrate scopes. What is the scalability of the method? (e.g. gram scale synthesis)
2. The molecular structures, reaction schemes, tables and experimental procedures were prepared flawlessly by the authors.
3. The NMR data of products as well as the synthesized substrates were found to be neat and clean with proper integration and chemical shift values.
4. The proposed reaction mechanism seems to be satisfactory. (Slight elaboration is recommended for the conversion from 6 to 7 & 8).
5. The formation of key intermediate is hydroxy(aryl)iodonium tosylate is characterized by using ESI and NMR analysis. In ESI method, the formation of intermediate is clearly visible from the m/z peak observed in the spectra. But, from 1H NMR, it is not clearly understood specially in case of step (iii) and step (iv). It is recommended to elaborate.
6. The authors claimed that as it is the first report of the intramolecular version of the catalytic oxidative cycloaddition of aldoximes using hypervalent iodine as a catalyst. However, authors may include the following work as a reference and may compare it with their methodology.
The thermal intramolecular cycloaddition of aldoxime was reported by Meshko and co-workers.Hassner, A.; Maurya, R.; Mesko, E. Tetrahedron Lett. 1988, 29, 5313.
7. It is recommended to cite following relevant articles along with references [40–61] related to importance of hypervalent iodine reagents in several synthetic organic reactions, including carbohydrate chemistry.
i) Chennaiah, A.; Vankar, Y. D. One-Step TEMPO-Catalyzed and Water-Mediated Stereoselective Conversion of Glycals into 2-Azido2-deoxysugars with a PIFA−Trimethylsilyl Azide Reagent System. Org. Lett. 2018, 20, 2611−2614
ii) Chennaiah, A.; Verma, A. K.; Vankar, Y. D. TEMPO-Catalyzed Oxidation of 3-O-Benzylated/Silylated Glycals to the Corresponding Enones Using a PIFA−Water Reagent System. J. Org. Chem. 2018, 83, 10535−10540
Overall, after addressing the points mentioned above, I recommend this article to publish in Molecules.
Author Response
June 11, 2022
Mr. Tory Tang
Assistant Editor
Molecules
Dear Mr. Tory Tang:
Thank you for thorough evaluation of our manuscript molecules- 1770717. We are submitting the revised manuscript that addresses all comments of the referees. Below you will find our response and the list of changes made in the manuscript; our response is in bold Italic.
Reviewer(s)' Comments to Author:
Referee 1
I have noted that there are several determinants missing such as "Heterocycles play A key role..." or "Isoxazole fragment is among THE most popular..." as well as some repetition like "stacking INTERACTIONS and hydrophilic INTERACTIONS", other sentences that do not sound quite right and some typos like "necassary". I advise the authors to review the whole text for better readibility.
We agree with the comment, we have fixed this.
I think that exposing first the objective of this study (third paragraph) and then, the role of heterocycles and cycloaddition (1st and 2nd paragraphs) would increase the cohesion of the article, since as a reader, I don't know why I should care about heterocycles until I read the 3rd paragraph. Or even better, to begin with a detailed explanation of what is isoxazole and why is it important to research on it rather than starting with heterocycles.
The authors are thankful the referee for the suggestion. We think that our introduction provides the most logical approach to the subject from the most general information to more specific details.
I cannot understand Scheme 3. Arrows point everywhere and I do not know which compounds enter the mechanism, which exit and which react together. Please, rework this scheme.
The authors are thankful the referee for the remark. We have fixed it.
Consider moving all the IR data in section 3.2 to the Supporting Information.
Thank you for your suggestions but we just followed the molecules Experimental Data rules, so we do not have a problem with that.
I would like to see a bit more depth into the mechanism development, since per the control experiments, I do not fully comprehend how can you reach the mechanism you proposed. More detail into why the authors think that each step represented is the actual reaction happening is needed.
The authors are thankful for the comment. Following to your advice we have added following sentances into the main text: “The key point of the reaction is generation of active hypervalent iodine species, which mediates an intermediate formation” (lines 100-102); “At the same time, we suggested that the active species can be formed with the 3-chlorobenzoic acid, which is produced during the oxidation of 2-iodobenzoic acid by m-CPBA.” (lines 112-114) and elaborated the sentence “Additionally, we have found that the reaction of protected oxime 5 under optimized con-ditions did not yield the desired product 3a (equation 4), and starting compound 5 was recovered from the reaction mixture. This experiment confirms a ligand exchange of hy-pervalent iodine species with aldoxime and subsequent nitrile oxide formation [62]” (lines 119-123)
Referee 2
Under the Results and discussion line 69 “thus 20 mol% of 2-iodobenzoic acid 2a is most suitable……” But at the foot note of Table-1 it shows 10 mol% of 2a.
We agree with the comment. We have fixed it, thank you.
The conversions in case of internal alkynes and alkenes (Table 2 3t-v) are interesting if it would be studied with a few more examples considering electronic effects.
We are thankful for the suggestion and nice comments. We will investigate this reaction using internal multiple bond compounds in the future.
Referee 3
The method seems to be very efficient with good yields and wide substrate scopes. What is the scalability of the method? (e.g. gram scale synthesis).
We are thankful for the suggestion. The method is applicable for transformation of 1g starting aldoxime 1a into 3a with the 96% yield. We added that result in the Table 2.
The proposed reaction mechanism seems to be satisfactory. (Slight elaboration is recommended for the conversion from 6 to 7 & 8).
We agree with the comment. Following to your advice we have changed the mechanism scheme in the manuscript.
The formation of key intermediate is hydroxy(aryl)iodonium tosylate is characterized by using ESI and NMR analysis. In ESI method, the formation of intermediate is clearly visible from the m/z peak observed in the spectra. But, from 1H NMR, it is not clearly understood specially in case of step (iii) and step (iv). It is recommended to elaborate.
We are thankful for the comment. The 1H NMR analysis of generation of active species was conducted for the reaction mixtures, thus the spectra look complex and, in addition, the active species are poorly soluble in CD2Cl2, which was used for the reaction and analysis.
The authors claimed that as it is the first report of the intramolecular version of the catalytic oxidative cycloaddition of aldoximes using hypervalent iodine as a catalyst. However, authors may include the following work as a reference and may compare it with their methodology.
The thermal intramolecular cycloaddition of aldoxime was reported by Meshko and co-workers.Hassner, A.; Maurya, R.; Mesko, E. Tetrahedron Lett. 1988, 29, 5313.
Thank you for the advice, we have added the comparison and corresponding references to the main text (lines 80-83) as follow: “In comparison with other approaches [37,40,70] to the synthesis of fused isoxazoles and isoxazolines our method is robust, afford comparable or higher yields of desired products, easy perfrom and does not require the use of excess of oxidant or heating for generation of intermediate - nitrile oxides.”
We look forward to hearing from you on this manuscript.
Sincerely yours,
Mekhman S. Yusubov
Professor Mekhman S. Yusubov (Corresponding Author)
The Tomsk Polytechnic University
Tomsk, Russia, 634050
E-mail: [email protected]
Round 2
Reviewer 2 Report
I recommend revised manuscript for publication